# Exploring psychiatrists' perspectives of working with patients with dissociative seizures in the UK healthcare system as part of the CODES trial: a qualitative study

Harriet Jordan,[1] Sarah Feehan,[1] Iain Perdue,[1] Joanna Murray,[2] Laura H Goldstein[1]

¹Department of Psychology, Institute of Psychiatry, Psychology and Neuroscience, King's College London, London, UK
²Health Service and Population Research, Institute of Psychiatry, Psychology and Neuroscience, King's College London, London, UK

**Correspondence to**
Professor Laura H Goldstein; laura.goldstein@kcl.ac.uk

## ABSTRACT

**Objective** There is currently limited research exploring healthcare professionals' (HCPs) experiences of working with patients with dissociative seizures (DS). Existing studies do not focus on the role of psychiatrists in treating this complex condition. The objective of this study was to gain an understanding of UK-based psychiatrists' experiences of the DS patient group. Against the backdrop of a UK-wide randomised controlled trial (RCT), the focus was broadened to encompass issues arising in everyday practice with the DS patient group.

**Design, participants and methods** A qualitative study using semistructured interviews was undertaken with 10 psychiatrists currently working with DS patients within the context of a large RCT investigating treatments for DS. Thematic analysis was used to identify key themes and subthemes.

**Setting** The psychiatrists were working in Liaison or Neuropsychiatry services in England.

**Results** The key themes identified were other HCPs' attitudes to DS and the challenges of the DS patient group. There is a clear knowledge gap regarding DS for many HCPs and other clinical services can be reluctant to take referrals for this patient group. Important challenges posed by this patient group included avoidance (of difficult emotions and help), alexithymia and interpersonal difficulties. Difficulties with alexithymia meant DS patients could struggle to identify triggers for their seizures and to express their emotions. Interpersonal difficulties raised included difficulties in attachment with both HCPs and family members.

**Conclusions** A knowledge gap for HCPs regarding DS has been identified and needs to be addressed to improve patient care. Given the complexity of the patient group and that clinicians from multiple disciplines will come into contact with DS patients, it is essential for any educational strategy to be implemented across the whole range of specialties, and to account for those already in practice as well as future trainees.

**Trial registration number** ISRCTN05681227; NCT02325544; Pre-results.

## Strengths and limitations of this study

► This study uniquely explores the experiences of psychiatrists providing healthcare to patients with dissociative seizures (DS).
► The findings have implications for guidance on interventions for people with DS, specifically in relation to epilepsy.
► The study has a small sample size of 10 psychiatrists. The psychiatrists were all currently working at healthcare centres across England.
► Psychiatrists working with DS patients in Scotland and Wales are not part of our sample.
► All the participants interviewed in this study were specialist psychiatrists with an interest and experience in working with patients with DS and, therefore, not representative of the population of psychiatrists more generally in the National Health Service across the UK.

## BACKGROUND

Dissociative seizures (DS) (often also referred to as Psychogenic Non-epileptic Seizures [PNES], Non-epileptic Attack Disorder or functional seizures) are similar in appearance to epileptic seizures without the abnormal neural activity. The incidence of DS is reported as approximately 4.9 per 100 000/per year,[1] with some estimates reaching as high as 50 per 100 000/per year.[2] DS are a common challenge in epilepsy centres worldwide,[3 4] with between 12% and 20% of patients referred for telemetry having coexisting or misdiagnosed DS.[5 6]

Quantitative research has indicated that there is a gap in the knowledge of healthcare professionals (HCPs) regarding DS[7] and that some HCPs have a negative attitude towards patients with DS, perceiving the seizures as being under their control and seeing DS

as untreatable.[8–10] Patients often describe feeling hopeless[11] and negative experiences with HCPs are frequently reported.[12] Previous research has found that clinicians, including general practitioners (GPs), have felt uncertainty in treating patients with DS[13 14] due to the lack of a substantial evidence-base for any one particular intervention. Similar results have also been found in the Danish paediatric setting, where clinicians also reported a lack of sufficient treatment options and a need for clinical guidance,[15] further demonstrating the impact of DS across age groups and cultures.

The CODES (COgnitive behavioural therapy vs standardised medical care for adults with Dissociative non-Epileptic Seizures) trial is the first sufficiently powered multicentre, pragmatic, parallel group randomised controlled trial (RCT) to investigate the effectiveness of any psychological therapy for patients with DS. CODES is evaluating the clinical effectiveness and cost-effectiveness of cognitive behavioural therapy (CBT) plus trial standardised medical care (SMC) compared with SMC alone.[16] Each patient recruited into the study was first seen by a neurologist and then referred on to a psychiatrist for assessment. This care pathway was not always normally available outside of the CODES trial in some areas of the UK within the context of the National Health Service (NHS).

When evaluating complex interventions such as those tested in the CODES trial, it can be difficult to capture effectiveness using only quantitative methods.[17] Within CODES, data on clinicians' views of the DS patient group and the intervention were felt to be most appropriately captured using qualitative methodology as it would allow participants to elaborate on their responses rather than being constrained by questionnaires. Research has found that combining quantitative and qualitative methods within a study overall provides essential insight into how and why an intervention is effective, if at all.[18 19] The purpose of this study was to gain an understanding of attitudes and beliefs among psychiatrists who had been part of the CODES trial and were experienced in working with patients with DS, with particular emphasis on psychiatrists' views of other HCPs' ability and willingness to work with DS patients in the context of the NHS.

## METHODS

### Study population

Ten participants were purposively selected from the 29 psychiatrists involved in the CODES RCT to encompass the geographical distribution of the CODES trial and the range of experience treating functional neurological disorders (FNDs), particularly DS. All participants were known to the wider CODES team prior to taking part in this qualitative study. CODES trial grant holders were excluded to avoid study design-related bias. The psychiatrists were based at nine different NHS Trust tertiary or secondary mental health services across England, with

one based in a specialist neurological hospital. Recruitment took place between June and September 2017.

### Data collection

Those selected were initially contacted via email by HJ and invited to take part. Half of those approached had a prior working relationship with HJ within the CODES trial. They were provided with an information sheet and a description of the aims of the project. If they were interested, a workplace-based face-to-face interview was scheduled at a time and date convenient for them. There were no refusals to participate. All interviews were conducted by HJ and recorded using an encrypted digital voice recorder to ensure data security and confidentiality. Interviews lasted between 41 and 96 min, covering the complete interview schedule.

Due to the nature of responses, it was not possible to determine the duration of responses solely covering the themes discussed in the current paper.

### Interview schedule

The interview schedule was developed by members of the CODES study team, of which all the authors were a part. The topics covered experiences of delivering the CODES SMC and involvement in the CODES RCT more generally (which will be reported elsewhere). In addition, topics covered the delivery of diagnosis, DS in the context of the NHS and the challenges of the patient group, which are the focus of this paper. Participants were encouraged to give examples where possible and probing techniques were used to explore responses and elicit further detail where necessary[20] (see online supplementary file 1). The interview began with a series of questions about aspects of the trial processes, which will be reported elsewhere. Although the topic guide focused on involvement in the CODES trial and distinguished this way of working from more general issues of working with patients with DS, the nature of responses meant that these topics often overlapped, and participants sometimes discussed issues relevant to multiple topics in a single answer. Participants readily elaborated with examples from their practice and experience outside of their involvement in the CODES RCT. During the stepwise coding and analysis of the interviews, several themes concerning the challenges of treating this patient group emerged. This inductive process inspired an analysis where the psychiatrists' accounts were used and contextualised through a specific focus on their views of other HCPs' attitudes to DS and their ability and willingness to work with patients with DS.

### Data analysis

The interviews were transcribed verbatim by members of the CODES research team. During the transcription process, the interview data were anonymised. Completed transcripts were checked by HJ against the original recordings to ensure accuracy. The semistructured interviews were analysed using thematic analysis.[21] This method was chosen rather than, for example, grounded theory, as our

aim was to understand participants' professional views and methods of working with patients with DS rather than to develop theory. Three of the completed transcripts were chosen at random and coded initially by HJ, SF and another member of the research team. Emerging findings and preliminary themes were discussed in team meetings. HJ and SF then coded all 10 transcripts independently, using the qualitative data analysis software NVivo V.11. NVivo allows the researcher to see how many interviews referred to a particular theme. Coding was done independently to allow for an organic and reflexive process. All content was grouped into categories to allow for the identification of patterns in the data. As each interview was analysed, new categories were added to the list and content was organised under each relevant category. Regular meetings were held to discuss agreements in coding and establish the parameters of each major theme. Major themes were established from the categories that contained the most substantial amount of data. Themes that had been identified by both coders were then combined, with subthemes being organised under the appropriate overarching theme. We believe saturation had been reached since, as the interviews progressed, it was clear no new major themes were being elicited.

### Patient and public involvement

The CODES trial has a number of service users (ie, individuals with DS or other relevant conditions) involved as members of its management committees, contributing to decisions about the running of the study and commenting on project outputs (eg, papers).

## RESULTS

Interviews from the 10 psychiatrists were analysed (see table 1 for the psychiatrists' demographic characteristics). In general, there was a consistent level of agreement among participants on most topics covered. This made it straightforward to identify main themes and clearly convey the conclusions drawn from the clinicians. Though the topic guide elicited a broad range of themes, for the purposes of this paper, we focused on those that had significant clinical implications. Other themes relating to the CODES trial will be described elsewhere. Two main clinically relevant overarching themes that emerged from the data were: (1) other HCPs and DS and (ii) psychiatrists' identified challenges of working with DS.

### Other HCPs and DS

#### HCPs ill-equipped to deal with DS

Psychiatrists reported that HCPs from other services often felt uncertain when dealing with DS patients or were not prepared to work with patients with functional neurological symptoms. Others felt that DS is a disorder that GPs should better understand. It was also reported that services would often contribute to the diagnostic confusion by continually mistaking DS for epilepsy, despite referrals stating otherwise;

**Table 1** Psychiatrists' self-reported demographic information

|  | N | % |
| --- | --- | --- |
| **Age** | | |
| 31–40 | 1 | 10 |
| 41–50 | 8 | 80 |
| 51–60 | 1 | 10 |
| **Gender** | | |
| Female | 5 | 50 |
| Male | 5 | 50 |
| **Location** | | |
| London | 6 | 60 |
| Rest of England | 4 | 40 |
| **Subspecialist accreditation** | | |
| Liaison psychiatry | 6 | 60 |
| Neuropsychiatry | 1 | 10 |
| Both | 3 | 30 |
| **Years of experience** | | |
| 11–15 | 5 | 50 |
| 16–20 | 2 | 20 |
| 21–25 | 1 | 10 |
| 26–30 | 2 | 20 |

They would come back saying well, look, this is epilepsy, they need to be seeing a neurologist, or people would end up back on anticonvulsants. (Psychiatrist 09, Female, Liaison Psychiatry)

The mention of seizures would often result in a panicked response from some primary care psychology services that meant patients could sense having something difficult to treat. Psychiatrists described patients often feeling other clinicians had not given a positive message about a DS diagnosis, with some GPs reportedly stating the need to be on an anticonvulsant simply at the mention of seizures. This continual reference to epilepsy by other professionals can have a negative impact on patients' progress;

They have said, oh this sounds … you have epilepsy. I say don't say that, you're not qualified to say that, you know, you do your job, uh because that one word would put patients (pause) back, by a year or two or ten sessions.(Psychiatrist 06, Male, Neuropsychiatry)

Psychiatrists would find that making DS referrals to psychology services would result in the referral being rejected unless patients had a comorbidity that psychologists felt they could treat;

So, if you send a referral saying this person has dissociative seizures, will you see them, they will return the referral, so you have to say, 'this person has dissociative seizures; however, they also have a very clear anxiety or panic disorder and that is what I would like you

to work on' and then they will accept it.(Psychiatrist 03, Female, Liaison Psychiatry)

This was reiterated throughout most of the interviews, with psychiatrists stating that local services would prefer to treat comorbidities rather than the DS themselves and where no comorbidity could be identified, services often rejected the referrals. The majority of the interviewees endorsed the view that psychiatrists were a key part of DS patient care. However, two of the 10 questioned whether it was necessary in all cases for a psychiatrist to be involved especially if the DS patient had no clear psychiatric comorbidities. This approach seemed to be influenced both by their usual practice and service pressures at the two trusts.

### The need for experience

One conclusion frequently drawn from psychiatrists' experiences with HCPs in other services was that, in order to diagnose and treat DS, the clinician needed to have a significant level of experience with the disorder and that treatment should be undertaken in a specialist setting;

I sincerely believe that … it's not a condition which anybody or everybody can deal with and I don't think it should be dealt with at IAPT level. (Psychiatrist 06, Male, Neuropsychiatry) (IAPT=Improving Access to Psychological Therapies services; the IAPT programme began in 2008 and aims to offer short-term evidence-based psychological treatments in particular for depression and anxiety in adults across England).

Delivering treatment in a specialist centre was described as not only helpful in terms of clinicians knowing how to work with DS, but would provide reassurance for patients that they were being seen by someone who is confident and experienced;

I think it's one of those conditions where seeing people who know what it is, know what to do with it even if they can't promise to get it better it reduces everybody's anxiety levels about it. (Psychiatrist 09, Female, Liaison Psychiatry)

This sense of needing experience was also reported as helpful in enabling professionals to acknowledge the amount that can often be unknown about the causes and triggers for DS and for helping the patient embrace that as well.

### Psychiatrists' identified challenges of working with DS
#### Avoidance

Avoidance was viewed as a key area of difficulty for the DS patient group and was noted to take a number of forms across 9/10 interviews. Examples of avoidance were given in the interviews but these fell into two broad categories: 'avoidance of help' and 'avoidance of emotions'. Avoidance of help included not reading information about DS even when handed this directly in an appointment and avoiding attending medical or therapy appointments. Avoidance of emotions included a desire to take

medication rather than deal with difficult feelings and blocking emotions;

Quite a lot of people may have blocked out so to speak, the more emotional side of things and try to get on with things…. If they start to go through a more open exploration of the issues this can be very emotionally distressing and suddenly their mood goes down … a lot of patients will have to go through that turbulence in order to come out on the other end having felt the issues, recognised them and dealt with it. (Psychiatrist 06, Male, Neuropsychiatry)

The seizures themselves were seen as potentially fitting into a pattern of avoidant behaviour as a defence against difficult emotions. Avoidant behaviour could present under the guise of other difficulties, such as a reluctance to travel to appointments.

Linked to the theme of avoidance was emotional literacy. This was commented on by 9/10 interviewees as a key difficulty for the DS patient group. It follows that if a person lacks awareness of their own emotions they would struggle to express these to others. This lack of emotional awareness could then impede the ability to make links between life events and feelings while in treatment;

No symptoms, happy go lucky kind of personality, I love my family, no trauma, no pain and those people are the hardest. (Psychiatrist 10, Female, Liaison Psychiatry)

Some of the most challenging of the DS patient group were those for whom no trigger for the seizures could be identified. Sometimes even when it was very clear to the clinician that there was a current stressor (such as caring for a gravely ill partner), patients with DS might deny this was the case. This seemed to lead to feelings of frustration for the psychiatrists as they viewed patients with no identified psychological trigger harder to treat successfully.

### Complex interpersonal relationships

Eight out of the 10 psychiatrists noted that a patient with DS may well struggle with relationships, both with people generally and within the clinician–patient relationship. This could be associated with difficulties in attachment, with the DS patient becoming over-attached and then not wanting to engage with any other clinician or be discharged;

She became quite attached; there were real attachment issues…. so, I only managed to discharge her as I said a few weeks ago … she didn't connect with that person (CODES CBT therapist) and she created a split between that person and me and she was like I just want to come and see you, can't I just come and see you every 2 weeks? (Psychiatrist 08, Female, Liaison Psychiatry)

Attempts at splitting were described as occurring not only between individual clinicians (as above) but also between services; for example, a specialist national service

being 'good' and all local services being 'bad'. This splitting and idealisation of one service or clinician could be accompanied by unrealistic expectations that the psychiatrist would continue to see them indefinitely. A partner being in the room throughout every appointment can mean the patient is less able to be open about how they are feeling and certainly less likely to discuss any current relationship difficulties. A codependent relationship may also impact on motivation to recover as some people with DS have social circumstances in which getting better may feel riskier than remaining unwell. A couple of the psychiatrists went further and conceptualised some examples of factitious behaviours in patients (eg, deliberately concealing medication and non-adherence from family and clinicians) as driven by a profound need to be looked after.

## DISCUSSION

We present, here, the experiences and views of the psychiatrists involved in an RCT for patients with DS. The characteristics of the nested group were similar to the clinicians involved in the trial as a whole in relation to age, gender, ethnicity and experience working with DS. Participants generally expressed concordant views across the range of interview questions, suggesting that issues surrounding DS are very apparent to the professionals working closely with this population. Views broadly support previous research that describes DS patients as a heterogeneous population with complex presentations and demanding HCP input,[8 10 21] as well as the need for an improvement in education and awareness of DS by HCPs.[22]

Participants believed that interventions by other HCPs at times made their own work with this patient group more difficult. They identified a knowledge gap surrounding DS among other HCPs. Previous research has found that HCPs from a variety of backgrounds often have very different perceptions regarding DS,[23] which seems to be a pervasive problem at all levels of health services in the UK, from GPs and primary care services to community mental health trusts and emergency departments. While we cannot report here on the perceptions of other HCPs involved in the CODES trial, what is perhaps most significant is the current participants' observation that this knowledge gap can at times have a detrimental effect on patients' prognosis, with one participant noting that the mention of epilepsy and antiepileptic drug (AED) treatment can set a patient's progress back significantly. Rawlings and Reuber's recent review[10] raised concerns not only about HCPs' DS knowledge gap but also negative attitudes towards the condition. It is of concern that negative attitudes towards the DS patient group may be created or reinforced by HCPs currently in practice when training junior staff.[10] Dworetzky[24] reported that epilepsy specialists in the USA when teaching junior staff about DS tend to focus on the cost of care and misuse of services caused by DS. Negative clinician attitudes towards DS patients themselves did not emerge as a theme here,

perhaps because our group of psychiatrists have chosen to work with this patient group.

Referring to epilepsy can be damaging to patients with DS as it contributes to diagnostic confusion in a number of ways. If they have been diagnosed with DS already, it may cause them to doubt whether the diagnosis is correct. It may initiate or strengthen a belief that they, in fact, have epilepsy, despite diagnostic evidence (eg, video-electroencephalography [vEEG] telemetry results) and clinical opinion to the contrary.[25] It also means that often DS patients are treated with potentially harmful AEDs with serious side effects despite them having no medical benefit and this can lead to serious iatrogenic harm.[26] It would be helpful for more specific epilepsy-related guidance to be developed for HCPs with regards to reducing AEDs and handling a misdiagnosis of epilepsy so as to avoid any setbacks in recovery. HCPs should be made aware of the clinical significance that simply mentioning epilepsy and AEDs can have on a patient and this should be highlighted in any future educational resources.

The International League Against Epilepsy PNES Task Force produced a special report describing the minimum requirements for a diagnosis of DS.[27] It would be beneficial for those working with DS to become familiar with these guidelines so that they can clearly convey their diagnostic reasoning to the patient and can discuss the features of DS and the characteristics distinguishing DS from epilepsy more confidently. In addition to this detailed account of the diagnosis, LaFrance et al[25] also produced a comprehensive overview of the management of DS patients, encompassing diagnosis, treatment and maintaining engagement. While it is argued that the management of DS requires expertise, the guidelines are accessible and can be used as a helpful tool for non-experts to familiarise themselves with the important elements for interacting with this patient group.

There were some discordant views surrounding which services and professionals should be able to work with DS. Some participants described the need for more specialist services to be made available and for psychiatric training to cover functional neurological symptoms in greater depth. However, some participants stressed the necessity for GPs and primary care clinicians to also become more familiar with DS. Despite this contrast, the consensus remains that more education and awareness are required for all HCPs regardless of their specialty. This supports previous research that has reported the need for more clinicians to be comfortable with treating all FNDs, particularly DS.[28] Encouragingly going forward even a brief training intervention for medical students and doctors' improved confidence levels and diagnostic accuracy when working with DS.[29 30]

To address this issue, better resources and educational materials need to be made available for those who are likely to be working with DS, both for future clinicians and clinicians currently in practice. Rommelfanger et al[22] described a level of 'professional isolation' often felt among care providers working with functional symptoms

due to a distinct lack of formalised training and reported on the need for a shift in priorities to support clinicians working with patients with FNDs. This could start as early as medical school but should also involve the development of sufficient resources to support the multidisciplinary approach that is often required to treat FNDs.

Avoidant behaviours could be divided into two categories of 'avoidance of help' and 'avoidance of emotions'. Linked to avoidance was the subtheme of emotional literacy, difficulty feeling and expressing emotions. These findings support previous quantitative research on alexithymia in patients with DS, where Bewley et al[31] found DS patients had significantly greater difficulty identifying feelings than healthy controls. More recently, Uliaszek et al[32] found significant emotion dysregulation among DS patients when compared with a control group. This may have important implications for future therapeutic developments, by incorporating elements of effective treatments for alexithymia from other treatment models (eg, mindfulness-based therapy for alexithymia).[33] Previous research also found evidence of experiential avoidance[34] and avoidant coping styles[35] among the DS patient population. Given these findings, it is important to establish what impact this avoidant behaviour may have on treatment outcomes and what can be done to mitigate any negative impacts. Some participants in the present study reported positive progress in openly identifying the patients' avoidant behaviour and providing the opportunity to address it constructively. Treatment approaches for DS should make provision for tackling avoidance directly.[16 36] In addition, for a subset of the patient group, no trigger for DS is identified and these patients tended to be particularly difficult to treat from the point of explaining the diagnosis onwards. Working with this complex patient group effectively clearly needs experience and knowledge of the condition.

Our study has a number of limitations. The study has a small sample size of 10 psychiatrists currently working at healthcare centres around England. Psychiatrists working in Scotland and Wales are not represented in this group. As participants were all involved in the CODES trial and knew the interviewer or other members of the research team, it is possible this influenced their responses. However, as the interview did not solely focus on the CODES trial, other aspects may be less likely to have been affected by prior working relationships.

It is possible that the DS patient sample in the CODES trial, who will have partly influenced the current psychiatrists' views, is not fully representative of the DS patient population in general. It is also likely that there is a self-selection bias for those presenting to psychiatrists as many DS patients may reject a psychiatric diagnosis and, thus, do not attend psychiatric appointments.[37] The disparity in services may mean that some patients simply do not have access to a specialist psychiatrist and are referred back to their GP.[38] We will report elsewhere an exploration of the views of patients with DS involved in the CODES trial to triangulate findings and maximise our understanding of working and living with DS. All participants in this current study were specialist psychiatrists and, therefore, not representative of the population of psychiatrists more generally in the NHS across the UK. However, due to the organisation of care in the NHS, DS patients are not usually seen by general psychiatrists. Therefore, our sample is representative of those clinicians who are most likely to provide direct clinical contact with the DS patient population. It is possible that talking about the challenges of the patient group would have led the respondents to think more about their perception of the difficulties of working with this patient group and may have influenced the nature of the emerging themes.

In terms of strengths, to the best of our knowledge, our study was the first qualitative study focusing solely on psychiatrists' perspectives of working with patients with DS. Prior published qualitative studies[39 40] interviewed HCPs from a variety of backgrounds. However, McMillan et al's[39] large sample (74 interviews) only included epilepsy staff, such as neurologists, EEG technicians and epilepsy nurses, and no mental healthcare providers. In du Toit and Pretorius's[40] study, only three of 15 people interviewed were psychiatrists. Given that DS is classified as a mental health disorder (DSM-5[41]) and a dissociative disorder (ICD-10[42]) and will predominantly be handled by mental health clinicians, it seems important to explore the views of those working with the DS population in the appropriate clinical context. In addition, it is likely that the current sample is more representative of the clinicians most likely to be working with DS in developed countries, and not limited to a military clinical environment.[39]

## CONCLUSION

Qualitative findings suggest that patients with DS are a complex and at times challenging population that requires intervention from experienced clinicians familiar with the condition. Significantly, intervention can be made more difficult if not provided in an informed and experienced manner. Our findings have important implications for medical and allied professional training with regards to FNDs in order for clinicians to be better equipped to recognise and handle the challenges that come with treating DS. It is also hoped that a greater evidence base for treatments for patients with DS will help to eradicate the variability within healthcare provision.

**Acknowledgements** The authors would like to thank the psychiatrists who were willing to be interviewed for this study given their busy schedules. The authors also thank Julie Read for her assistance with the project.

**Contributors** HJ, IP, JM and LHG contributed to the development of the topic guide. HJ carried out all the data collection. HJ and SF transcribed the data, carried out the analysis and led the writing of the paper. JM and IP reviewed drafts of the paper. LHG oversaw the project, contributed to the writing of the paper and reviewed successive drafts. All authors approved the final version of the submitted text.

**Funding** This paper describes independent research funded by the National Institute for Health Research (NIHR) (Health Technology Assessment programme, 12/26/01, COgnitive behavioural therapy vs standardised medical care for adults with Dissociative non-Epileptic Seizures: a multicentre randomised controlled trial [CODES]). LHG receives salary support from the NIHR Maudsley Biomedical

Research Centre at the South London and Maudsley NHS Foundation Trust and King's College London.

**Disclaimer** The views expressed in this publication are those of the authors and not necessarily those of the NHS, the NIHR or the Department of Health and Social Care.

**Competing interests** LHG and JM report the grant from the NIHR HTA for the conduct of the study. None of the other authors have competing interests to declare.

**Patient consent for publication** Not required.

**Ethics approval** The study was approved by the NHS Health Research Authority, London—Camberwell St Giles Research Ethics Committee (reference number 13/LO/1595). Written informed consent was obtained from all study participants. Study registration: Current Controlled Trials: ISRCTN05681227; ClinicalTrials.gov: NCT02325544.

**Provenance and peer review** Not commissioned; externally peer reviewed.

**Data sharing statement** Contact the corresponding author with queries.

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
