## [Reviewer comments · BMJ Open]

ARTICLE DETAILS

TITLE (PROVISIONAL)	Exploring psychiatrists' perspectives of working with patients with dissociative seizures in the UK healthcare system as part of the CODES trial – A qualitative study
AUTHORS	Jordan, Harriet; Feehan, Sarah; Perdue, Iain; Murray, Joanna; Goldstein, Laura

VERSION 1 - REVIEW

REVIEWER	Laura Scévola Ramos Mejia Hospital, Buenos Aires, Argentina
REVIEW RETURNED	29-Sep-2018

GENERAL COMMENTS	This qualitative study correctly reflects what we see in our daily practice the psychiatrists that work on this issue (in my case psychogenic non epileptic seizures).
--

REVIEWER	Charlotte Ulrikka Rask Aarhus University Hospital, Denmark
REVIEW RETURNED	08-Oct-2018

GENERAL COMMENTS	The paper presents results from a qualitative study on psychiatrists' perspectives of their clinical work with patients with dissociative seizures. The data stem from semi-structured interviews with 10 psychiatrists, all taking part as assessors in the large ongoing RCT (CODES trial) investigating treatments for DS. Thematic analysis is used to identify key themes and subthemes. Overall, the paper is well-written and the results are clearly presented and nicely discussed – also in relation to future implications and perspectives. There are a few features which could be refined in order to improve the paper even more. Below follow more detailed comments which I would like the authors to address. Title I suggest the authors already make a reference to the CODES trial in the title. Introduction This section is clearly written and provides the rationale for the study as research in this area is lacking. The case could be made even stronger by mentioning that the lack of knowledge of HCPs regarding DS is a general issue across age groups, i.e. the same problem has been reported in the pediatric literature, for example
---

	in the recent paper by Nielsen ES et al, European Journal of Pediatric Neurology 2018 Sep;22(5):774-781. I don't find the purpose described in the end of this section, i.e. that there will be a particular emphasis on psychiatrists' ability and willingness to work with DS fits what is later presented in the result section. The data seem more related to the psychiatrist' view on other HCP's abilities and willingness to work with this patient group. I therefore suggest reformulating this part. Method Data collection: It follows from the COREQ checklist that there were no refusals with regard to participation which is a strength of the study. I suggest adding this information here in this section. The interview schedule does not seem to particular fit the described purpose of this study but mainly to explore the psychiatrists' experiences of delivering CODES SMC and involvement in the CODES RCT (it is stated these data will be presented in a separate paper). One could get the impression that during the overall data analysis, themes concerning the challenges of treating this patient group emerged which gave the idea for the current paper. Please elaborate and state more clearly if this was the case? Results The results are very well presented and with interesting points of clinical value. Discussion Again, this is a well-written section. However, it could be relevant to discuss in more detail the potential implications of the interview persons all being dedicated clinicians with regard to treating these patients. Therefore it is likely that they do not represent the perspectives of psychiatrists' in general. Final comment I would like to congratulate the authors with the nice CODES study. This is really a huge task to take on and it will provide us with very important knowledge on how to better treat these complex patients. The combination of an qualitative and quantitative approach really increases the overall quality of this important study. I look forward to learn more about the results.
--	---

REVIEWER	Joseph Cerimele MD MPH Assistant Professor Department of Psychiatry & Behavioral Sciences University of Washington School of Medicine 1959 NE Pacific Street, Box 356560 Seattle, WA 98198-6560 United States
REVIEW RETURNED	19-Nov-2018

GENERAL COMMENTS	Introduction 1) The third paragraph of the Introduction describes how the clinical work in the trial may have differed from usual clinical work. This information seems more relevant to the Discussion section – perhaps this distinction in clinical work flow informed responses to the interview questions – rather than in the Introduction. Moving
--

this text to the Discussion could open space in the Introduction for additional justification for pursuing the study, i.e. why would qualitative data collection answer the question.

Methods

1) Participants were known to the investigators prior to participation in this study. How might an existing relationship influence participant consent to participate in research or responses to qualitative questions? COREQ item 13 states no refusals to participation – my opinion is that should be included in the manuscript.

2) Did the reported interview duration include the first section of the interview on aspects of the trial processes (not reported here), and do the investigators believe the initial part of the interview influenced the second part of the interview? If so, what was the estimated interview duration for data reported in this manuscript?

3) Was a conceptual model developed to inform interview guide development, or to guide the thematic analysis? Why did the investigators choose thematic analysis compared to other methods of analysis of individual interview data such as grounded theory? Additional description of why thematic analysis was used would inform readers.

4) I am unfamiliar with service user involvement or review – will the authors add one additional clarifying phrase or sentence to describe this in greater detail?

5) The analysis section describes adding “new themes to the list” during analysis though only two themes are reported in the Results. Will the authors clarify this point? Some investigators use categories during analysis.

Results

1) The first paragraph describes a high level agreement which is not a usual way (that I have seen) to report results of qualitative data. I more commonly see methods/results of collecting and analyzing data until saturation - meaning no new information is being uncovered in collection or analysis. I wonder if an adequate amount of data was collected since the views/opinions of participants were reportedly homogeneous. Alternatively, would the authors describe additional detail about having reached saturation or not. The COREQ item 22 on saturation states N/A.

2) In the description of the first theme, the authors have at times reported quantitative data such as how many participants responded in such a way. Though perhaps straightforward counting, the authors had not reported in the methods how these results were collected. If the investigators did tally types of comments then that should be reported as a method, along with which types of comments were counted.

3) Example quotes from theme 1 regarding need for specialization may be just as applicable for theme 2 – i.e. specialization among psychiatrists – will the authors describe their rationale in greater detail for considering data on specialization to apply to other clinicians and not to psychiatrists in this study? If need for

	specialization was a prominent part of the data reflected in coding, would it warrant its own theme? 4) The psychiatrist-identified challenges seem more related to their impressions of individuals' with DS's challenges i.e. avoidance. I was surprised that the text on Theme 2 did not include data on the psychiatrists' personal experiences with clinical challenges – the included data report on their observations of patients. Currently, theme 2 appears to describe the phenomenology of DS, symptoms and potential symptom severity, defenses such as splitting, and associated psychosocial stressors. I wonder if symptom severity/stressors/challenging clinical scenarios is more of a theme, which could go along with another potential theme of perceived need for specialization in clinicians. Discussion 1) Paragraphs 2 includes a sentence that negative clinician attitudes did not emerge from the data, though the reported data on current theme 1 seem to reflect negative clinician attitudes towards other clinicians regarding treatment decisions.
--	--

REVIEWER	Benjamin Tolchin Yale University School of Medicine VA Connecticut Healthcare System
REVIEW RETURNED	20-Nov-2018

GENERAL COMMENTS	The authors report on a qualitative study of the attitudes and beliefs of psychiatrists participating in the CODES trial (randomized trial of CBT for the treatment of dissociative non-epileptic seizures or DS). They conducted semi-structured interviews with 10 of the 27 psychiatrists participating in CODES, practicing at NHS sites across England, then performed thematic analysis, coding transcripts for themes in a reflexive process. Major themes included that: other health care providers are often ill-equipped to deal with DS, that providers require significant experience/expertise to deal effectively with DS, that patients are highly avoidant of help and emotions, and that patients often struggle with relationships. The study is novel and clinically important in that – despite the critical role of psychiatrists in the evaluation and treatment of DS -- there are relatively limited qualitative data about psychiatrists attitudes and beliefs about DS. There are undeniably numerous obstacles to the treatment of patients with PNES, and the perspective of psychiatrists is critical in understanding and overcoming these obstacles. This study gives an in-depth picture of the attitude and beliefs of one group of English psychiatrists, and draws important recommendations from these data. Having said that, I do have some (relatively minor) reservations about the degree to which these psychiatrists are representative and the acceptance of the psychiatrists' views in the absence of other perspectives. 1. All participating psychiatrists are involved in one of the first multisite RCTs of CBT for the treatment of PNES. It is not obvious to me that these participants would be representative of English
--

psychiatrists. I would think they would be disproportionately subspecialists, working in academic medical centers, geographically biased toward large cities, and possibly therefore sharing similar attitudes and beliefs. Table 1 only strengthens my concerns, revealing that participants are 80% aged 41-50, 60% located in a single city (London), and 100% subspecialists (liaison, neuropsychiatry, or both). The fact that each of the identified themes shows up in 8 or 9 of the 10 interviews further strengthens my suspicion that we are getting the perspective of a relatively homogenous and tightly knit group. In discussing their limitations, the authors correctly state that theirs is a small sample size and speculate that it is possible that the DS patient sample in the CODES trial is not fully representative of the DS patient population in general. I would recommend additionally stating in the limitations that the psychiatrists interviewed are likely not representative of English psychiatrists generally (perhaps stating that they are rather representative of a particularly experienced and expert subspecialist group). Alternatively, if you have reason to think that these psychiatrists are representative of all English psychiatrists despite their apparently disproportionate geographic and subspecialty training, I think the reasons for that belief need to be spelled out.

2. Consider listing the practice location for the 10 respondents in table 1 (academic medical center, community clinic, etc.). I am concerned that in addition to their other similarities, most or all also share a similar practice setting.

3. The psychiatrists' criticism of GPs, psychologists, and emergency medicine providers raises questions in my mind about the lack of perspectives from these other providers. Of course the authors cannot and should not provide the perspectives of all health care providers in this manuscript. Still, it is one thing to present the study results as the perspective of (a small and very subspecialized and expert group of) psychiatrists. It is another to present the results as a complete and accurate picture of the problems in the treatment of DS. Yet the authors seem to be accepting the psychiatrists' perspective as essentially accurate when they state that HCPs from different backgrounds having different perceptions regarding DS is "a pervasive problem at all levels of health services in the UK, from GPs and primary care services to Community Mental Health Trusts and Emergency Departments."(page 11, lines 34-38). This seems like an excessive claim in the absence of any data from GPs, primary care providers, emergency department providers, etc. The conclusion should be toned down to make it clear that this is the perception of a specific group of subspecialist psychiatrists. It would be reasonable to state that the authors' findings support the conclusions of Rawling and Reuber's systematic review of HCP's perceptions of DS (*Epilepsia* 2018;59(6)1109-23, which the authors do reference), as this review does include studies of GPs, psychologists, emergency physicians, and other health care providers. The absence of GP/PCP/psychologist/emergency medicine perspective in the current data, and the possibility that these perspectives might be diverge in important ways from the respondents' views, might be addressed in limitations. Alternatively, in the discussion, the authors might site studies that actually did collect data from these other health care provider groups, and show how they agreed or disagreed with the current study's psychiatrists.

	4. The topics covered by interviewers in the semi-structured interviews included the challenges of the patient group, which may have biased respondents toward their second overarching theme (“psychiatrists’ identified challenges of working with DS”).
--	--

VERSION 1 – AUTHOR RESPONSE

RESPONSES TO REVIEWERS’ COMMENTS

Reviewer: 1

Reviewer Name: Laura Scévola

Institution and Country: Ramos Mejia Hospital, Buenos Aires, Argentina Please state any competing interests or state ‘None declared’: None declared

[Please leave your comments for the authors below]

This qualitative study correctly reflects what we see in our daily practice the psychiatrists that work on this issue (in my case psychogenic non-epileptic seizures).

Our Response:

Thank you for taking the time to review our paper. It is appreciated.

Reviewer: 2

Reviewer Name: Charlotte Ulrikka Rask

Institution and Country: Aarhus University Hospital, Denmark Please state any competing interests or state ‘None declared’: None declared

[Please leave your comments for the authors below Reviewer comments]

a) The paper presents results from a qualitative study on psychiatrists’ perspectives of their clinical work with patients with dissociative seizures. The data stem from semi-structured interviews with 10 psychiatrists, all taking part as assessors in the large ongoing RCT (CODES trial) investigating treatments for DS. Thematic analysis is used to identify key themes and subthemes.

Overall, the paper is well-written and the results are clearly presented and nicely discussed – also in relation to future implications and perspectives.

There are a few features which could be refined in order to improve the paper even more. Below follow more detailed comments which I would like the authors to address.

Title

I suggest the authors already make a reference to the CODES trial in the title.

Our Response:

We have now made reference to the CODES trial in the title.

b) Introduction

This section is clearly written and provides the rationale for the study as research in this area is lacking. The case could be made even stronger by mentioning that the lack of knowledge of HCPs regarding DS is a general issue across age groups, i.e. the same problem has been reported in the pediatric literature, for example in the recent paper by Nielsen ES et al, European Journal of Pediatric Neurology 2018 Sep;22(5):774-781.

Our Response:

Thank you. We have added a comment to this effect and this reference at the end of the second paragraph in the Background (Introduction) section.

c) I don't find the purpose described in the end of this section, i.e. that there will be a particular emphasis on psychiatrists' ability and willingness to work with DS fits what is later presented in the result section. The data seem more related to the psychiatrist' view on other HCP's abilities and willingness to work with this patient group. I therefore suggest reformulating this part.

Our Response:

We have clarified the emphasis of the paper at the very end of the Background section.

d) Method

Data collection: It follows from the COREQ checklist that there were no refusals with regard to participation which is a strength of the study. I suggest adding this information here in this section.

Our Response:

We have added this information to the Data Collection section.

e) The interview schedule does not seem to particular fit the described purpose of this study but mainly to explore the psychiatrists' experiences of delivering CODES SMC and involvement in the CODES RCT (it is stated these data will be presented in a separate paper). One could get the impression that during the overall data analysis, themes concerning the challenges of treating this patient group emerged which gave the idea for the current paper. Please elaborate and state more clearly if this was the case?

Our Response:

The interview schedule covered both the experience of delivering CODES SMC and involvement in the CODES RCT (information about the latter will be presented elsewhere) and the challenges of treating this patient group (e.g. Section 3 of the Interview Schedule). However, due to the nature of the interview (i.e. that participants were encouraged to elaborate and use examples where possible), answers often covered both providing CODES SMC and challenges of treating this patient group in their usual practice; hence separate themes emerged from the same question or topic. We have now addressed this issue in the Interview Schedule section.

f) Results

The results are very well presented and with interesting points of clinical value.

Our Response:

We appreciate this comment.

g) Discussion

Again, this is a well-written section. However, it could be relevant to discuss in more detail the potential implications of the interview persons all being dedicated clinicians with regard to treating these patients. Therefore it is likely that they do not represent the perspectives of psychiatrists' in general.

Our Response:

We have added a brief statement to our Limitations section acknowledging the lack of general psychiatrists in our sample but also identifying that general psychiatrists are unlikely to come across DS patients in the NHS due to the organisation of NHS services.

h) Final comment

I would like to congratulate the authors with the nice CODES study. This is really a huge task to take on and it will provide us with very important knowledge on how to better treat these complex patients. The combination of an qualitative and quantitative approach really increases the overall quality of this important study. I look forward to learn more about the results.

Our Response:

We greatly appreciate this comment.

Reviewer: 3

Reviewer Name: Joseph Cerimele MD MPH

Institution and Country: Assistant Professor Department of Psychiatry & Behavioral Sciences
University of Washington School of Medicine

1959 NE Pacific Street, Box 356560

Seattle, WA

98198-6560

United States

Please state any competing interests or state 'None declared': None declared

Please leave your comments for the authors below BMJ Open Review 2018-026493

a) Introduction

1) The third paragraph of the Introduction describes how the clinical work in the trial may have differed from usual clinical work. This information seems more relevant to the Discussion section – perhaps this distinction in clinical work flow informed responses to the interview questions – rather than in the Introduction.

Our Response:

Thank you for your comment. We feel it is important for the context of the paper to set the scene and indicate early on that the practise required of clinicians within the CODES RCT differed (to a varying extent) from routine clinical work. This “setting the scene” also highlights the lack of a standard care pathway for people with DS in the UK. We have, however, slightly reworded this aspect as the wording was not entirely clear.

2) Moving this text to the Discussion could open space in the Introduction for additional justification for pursuing the study, i.e. why would qualitative data collection answer the question.

Our Response:

In the Background section, 4th paragraph, we have added in an additional two sentences to help justify the use of qualitative methodology.

b) Methods

1) Participants were known to the investigators prior to participation in this study. How might an existing relationship influence participant consent to participate in research or responses to qualitative questions?

Our Response:

In the Data Collection section, we have added in a short sentence to make clear half of the participants and interviewer (HJ) already had a working relationship within the CODES study.

We cannot rule out the possibility that being involved in CODES and knowing members of the research team influenced Psychiatrists' decisions to take part and their responses within the interviews.

However, they were aware that participation was voluntary and the overall impression from the interviews is that they were open and honest, highlighting both positives and negatives of involvement in CODES. The interviews did not focus solely on CODES, so these aspects may be less likely to be influenced by prior relationships. A comment on this has been added into the limitations section.

2) COREQ item 13 states no refusals to participation – my opinion is that should be included in the manuscript.

Our Response:

In the Data Collection section we have now noted that we had no refusals to participate.

3) Did the reported interview duration include the first section of the interview on aspects of the trial processes (not reported here), and do the investigators believe the initial part of the interview influenced the second part of the interview? If so, what was the estimated interview duration for data reported in this manuscript?

Our Response:

Due to the nature of responses, it was not possible to determine the duration of responses solely covering the themes discussed in the current paper. The duration times reported are the total interview length, as we have now made clear in the Data Collection section.

Because the interview schedule moves between CODES and non-CODES questions throughout it does not divide neatly into halves, so one half influencing the next is thought to be unlikely.

c) Was a conceptual model developed to inform interview guide development, or to guide the thematic analysis? Why did the investigators choose thematic analysis compared to other methods of analysis of individual interview data such as grounded theory? Additional description of why thematic analysis was used would inform readers.

Our Response:

We have now given our rationale for using thematic analysis rather than grounded theory in the Data Analysis section

d) I am unfamiliar with service user involvement or review – will the authors add one additional clarifying phrase or sentence to describe this in greater detail?

Our Response:

We may have initially attempted to be too succinct in our writing and have now attempted to clarify this in the section on Patient and Public Involvement. Of potential interest to the reviewer is the following paper <https://www.ncbi.nlm.nih.gov/pubmed/25684242> although we have not added this to the paper.

f) The analysis section describes adding “new themes to the list” during analysis though only two themes are reported in the Results. Will the authors clarify this point? Some investigators use categories during analysis.

Our Response:

Thank you. In the Data Analysis section, we have amended the phrase to say categories instead and clarified how these were developed into themes.

g) Results

1) The first paragraph describes a high-level agreement which is not a usual way (that I have seen) to report results of qualitative data. I more commonly see methods/results of collecting and analyzing data until saturation - meaning no new information is being uncovered in collection or analysis. I wonder if an adequate amount of data was collected since the views/opinions of participants were reportedly homogeneous.

Alternatively, would the authors describe additional detail about having reached saturation or not. The COREQ item 22 on saturation states N/A.

Our Response:

The wording of the first paragraph of the Results section has been changed to say consistent level of agreement as these themes were strongly endorsed across the interviews.

Although many of the same sub-themes were raised across the interviews as being important we have commented on two areas of divergence: a) in the section “Other Healthcare Professionals and DS” regarding whether it is always necessary for a psychiatrist to see a patient with DS only and b) “Complex Interpersonal Relationships” on factitious behaviours.

We believe saturation had been reached since, as the interviews progressed, it was clear no new major themes were being elicited.

We apologise for the apparent error in the COREC checklist which we have now amended.

2) In the description of the first theme, the authors have at times reported quantitative data such as how many participants responded in such a way. Though perhaps straightforward counting, the authors had not reported in the methods how these results were collected. If the investigators did tally types of comments then that should be reported as a method, along with which types of comments were counted.

Our Response:

NVIVO software allows the user to see how many interview quotes have been coded under each theme and how many interviews touched on each theme. So, it becomes easy to see which themes came up most frequently across the interviews.

We have added a sentence in the Methods section, Data Analysis to make say NVIVO makes this information readily available.

3) Example quotes from theme 1 regarding need for specialization may be just as applicable for theme 2 – i.e. specialization among psychiatrists – will the authors describe their rationale in greater detail for considering data on specialization to apply to other clinicians and not to psychiatrists in this study? If need for specialization was a prominent part of the data reflected in coding, would it warrant its own theme?

Our Response:

Our Data was divided into two overarching or major themes 1) Other HCP's and DS and 2) Psychiatrists' Identified Challenges of Working with DS. This reflected the manner in which the information was provided by the interviewees. We do appreciate that an alternative coding might have been to have an overall theme of specialisation but we do not feel that such recoding at this stage would yield a substantially different interpretation of the data.

4) The psychiatrist-identified challenges seem more related to their impressions of individuals' with DS's challenges i.e. avoidance. I was surprised that the text on Theme 2 did not include data on the psychiatrists' personal experiences with clinical challenges – the included data report on their observations of patients. Currently, theme 2 appears to describe the phenomenology of DS, symptoms and potential symptom severity, defenses such as splitting, and associated psychosocial stressors. I wonder if symptom severity/stressors/challenging clinical scenarios is more of a theme, which could go along with another potential theme of perceived need for specialization in clinicians.

Our Response:

With respect to Major Theme 2 “Psychiatrists’ Identified Challenges of Working with DS” we have endeavoured to represent the data in the way that the Psychiatrists themselves identified the challenges. The sub-themes covered “avoidance” (and within “avoidance” “emotional literacy” and “lack of an identified trigger”) “complex interpersonal relationships” (and within this “attachment”). These can be both features of the DS patient group and clinical challenges.

On page 11 we link, as the Psychiatrists did themselves, struggles with emotional literacy, lack of an identified trigger for the DS and susceptibility to treatment. This subset of DS patients were clearly being viewed as difficult to treat effectively, leading to feelings of frustration for the Psychiatrists themselves.

Perceived need for specialization in clinicians

The psychiatrists we interviewed are themselves specialists with a particular interest in the DS patient group. Their concern was that other healthcare professionals frequently lack awareness even that the condition DS exists.

We hope the text is sufficiently clear in this respect.

h) Discussion

1) Paragraphs 2 includes a sentence that negative clinician attitudes did not emerge from the data, though the reported data on current theme 1 seem to reflect negative clinician attitudes towards other clinicians regarding treatment decisions.

Our Response:

The negative attitudes flagged up as a concern in Rawlings and Reuber's (2018) review paper and elsewhere in the literature are referring to negative attitudes of HCP's towards people with DS. We have added to the relevant sentence to clarify our intended meaning.

Reviewer: 4

Reviewer Name: Benjamin Tolchin

Institution and Country: Yale University School of Medicine VA Connecticut Healthcare System
Please state any competing interests or state 'None declared': None declared

a)The authors report on a qualitative study of the attitudes and beliefs of psychiatrists participating in the CODES trial (randomized trial of CBT for the treatment of dissociative nonepileptic seizures or DS). They conducted semi-structured interviews with 10 of the 27 psychiatrists participating in CODES, practicing at NHS sites across England, then performed thematic analysis, coding transcripts for themes in a reflexive process. Major themes included that: other health care providers are often ill-equipped to deal with DS, that providers require significant experience/expertise to deal effectively with DS, that patients are highly avoidant of help and emotions, and that patients often struggle with relationships.

The study is novel and clinically important in that – despite the critical role of psychiatrists in the evaluation and treatment of DS -- there are relatively limited qualitative data about psychiatrists' attitudes and beliefs about DS. There are undeniably numerous obstacles to the treatment of patients with PNES, and the perspective of psychiatrists is critical in understanding and overcoming these obstacles. This study gives an in-depth picture of the attitude and beliefs of one group of English psychiatrists and draws important recommendations from these data.

Having said that, I do have some (relatively minor) reservations about the degree to which these psychiatrists are representative and the acceptance of the psychiatrists' views in the absence of other perspectives.

1. All participating psychiatrists are involved in one of the first multisite RCTs of CBT for the treatment of PNES. It is not obvious to me that these participants would be representative of English psychiatrists. I would think they would be disproportionately subspecialists, working in academic medical centers, geographically biased toward large cities, and possibly therefore sharing similar attitudes and beliefs.

Table 1 only strengthens my concerns, revealing that participants are 80% aged 41-50, 60% located in a single city (London), and 100% subspecialists (liaison, neuropsychiatry, or both). The fact that each of the identified themes shows up in 8 or 9 of

the 10 interviews further strengthens my suspicion that we are getting the perspective of a relatively homogenous and tightly knit group.

Our Response:

We thank the reviewer for these comments. It is correct that in the UK specialist services for DS patients are predominantly located in large cities. Six of our participant psychiatrists are London based (though cover large catchment areas outside London) and the remaining 4 are based in large cities around the UK, as indicated in Table 1. We have made a general comment in the Study Population section on the nature of the NHS services in which they work and commented again in the limitations on the nature of the sample. We have already commented that whilst the current sample is not representative of general psychiatrists in the UK, DS patients are unlikely to be seen by general psychiatrists.

b) In discussing their limitations, the authors correctly state that theirs is a small sample size and speculate that it is possible that the DS patient sample in the CODES trial is not fully representative of the DS patient population in general. I would recommend additionally stating in the limitations that the psychiatrists interviewed are likely not representative of English psychiatrists generally (perhaps stating that they are rather representative of a particularly experienced and expert subspecialist group).

Alternatively, if you have reason to think that these psychiatrists are representative of all

English psychiatrists despite their apparently disproportionate geographic and subspecialty

training, I think the reasons for that belief need to be spelled out.

Our Response:

We have added to our limitations section to clarify this is an experienced and specialist group of Psychiatrists who may not be representative of the general psychiatry population across the UK. However, there is no standard care pathway for DS patients in the UK and they are rarely seen by general psychiatrists unless presenting primarily with another co-morbidity. So, our sample is representative of the Psychiatrists most likely to have direct clinical contact with the DS patient population.

c). Consider listing the practice location for the 10 respondents in table 1 (academic medical center, community clinic, etc.). I am concerned that in addition to their other similarities, most or all also share a similar practice setting.

Our Response:

We have added a short sentence to the study population paragraph within the Methods (Study Population) section to describe the practice settings of the Psychiatrists. It is also worth noting that some of the psychiatrists interviewed have more than one work base.

We are mindful here of not providing too much information that might identify our participants.

d) The psychiatrists' criticism of GPs, psychologists, and emergency medicine providers raises questions in my mind about the lack of perspectives from these other providers. Of course the authors cannot and should not provide the perspectives of all health care providers in this manuscript. Still, it is one thing to present the study results as the perspective of (a small and very subspecialized and expert group of) psychiatrists. It is another to present the results as a complete and accurate picture of the problems in the treatment of DS. Yet the authors seem to be accepting the psychiatrists' perspective as essentially accurate when they state that HCPs from different backgrounds having

different perceptions regarding DS is “a pervasive problem at all levels of health services in the UK, from GPs and primary care services to Community Mental Health Trusts and Emergency Departments.”(page 11, lines 34-38). This seems like an excessive claim in the absence of any data from GPs, primary care providers, emergency department providers, etc. The conclusion should be toned down to make it clear that this is the perception of a specific group of subspecialist psychiatrists. It would be reasonable to state that the authors’ findings support the conclusions of Rawling and Reuber’s systematic review of

HCP’s perceptions of DS (Epilepsia 2018;59(6)1109-23, which the authors do reference), as this review does include studies of GPs, psychologists, emergency physicians, and other health care providers. The absence of GP/PCP/psychologist/emergency medicine perspective in the current data, and the possibility that these perspectives might be diverge in important ways from the respondents’ views, might be addressed in limitations. Alternatively, in the discussion, the authors might site studies that actually did collect data from these other health care provider groups, and show how they agreed or disagreed with the current study’s psychiatrists.

Our Response:

We have attempted to clarify our text (Discussion section) and to indicate that we are not able to present the views of other HCPs here but are referring solely to the views of our currently-interviewed psychiatrists.

e) The topics covered by interviewers in the semi-structured interviews included the challenges of the patient group, which may have biased respondents toward their second overarching theme (“psychiatrists’ identified challenges of working with DS”).

Our Response:

We have added a statement to reflect this at the end of the limitations section.

VERSION 2 – REVIEW

REVIEWER	Charlotte Ulrikka Rask Aarhus University Hospital, Denmark
REVIEW RETURNED	21-Jan-2019

GENERAL COMMENTS	Thank you for the opportunity to review this revised manuscript where the authors have fairly comprehensive addressed the concerns and points of clarification from the first review. OBS It would have been helpful for the review process if all the changes had been clearly marked in the revised manuscript. I have a few minor comments left:  1. The article summary should more clearly reflect the limitation on the nature of the study sample (i.e. the representativeness of the participating psychiatrists). 2. I don't find the response regarding how the used interview schedule came about to cover one of the described topics of the current study, i.e. other health care professionals and DS, is quite clear - nor well described in the manuscript (page 7 line 8-9: "...and it became apparent during data analysis that the themes covered warranted the focus of the current paper.").
--

	Suggestion: could it be formulated like this? "During the stepwise coding and analysis of the interviews, several themes concerning the challenges of treating this patient group emerged. This inductive process inspired an analysis where the psychiatrists' accounts were used and contextualised through a specific focus on their views of other HCP's attitudes to DS and ability and willingness to work with patients suffering from DS."
--	--

REVIEWER	Joseph Cerimele MD University of Washington School of Medicine 1959 NE Pacific Street, Box 356560, Seattle, WA 98195-6560 United States of America
REVIEW RETURNED	14-Jan-2019

GENERAL COMMENTS	The authors have provided informative responses to reviewer comments, and have revised the manuscript. I have no additional comments.
---

REVIEWER	Benjamin Tolchin Yale University School of Medicine, USA
REVIEW RETURNED	14-Jan-2019

GENERAL COMMENTS	I am satisfied with the authors' response to the prior round of review. My previously expressed concerns have been addressed.
---

VERSION 2 – AUTHOR RESPONSE

RESPONSES TO REVIEWERS' COMMENTS FOR SECOND REVISION

We thank reviewers 2-4 for taking the time to re-review the paper.

Comments

1. "The article summary should more clearly reflect the limitation on the nature of the study sample (i.e. the representativeness of the participating psychiatrists)."

Our response

We have added the following point to the Article Summary

- All the participants interviewed in this study were specialist psychiatrists with an interest and experience in working with patients with DS and therefore not representative of the population of psychiatrists more generally in the National Health Service across the UK.

2. "I don't find the response regarding how the used interview schedule came about to cover one of the described topics of the current study, i.e. other health care professionals and DS, is quite clear - nor well described in the manuscript (page 7 line 8-9: "...and it became apparent during data analysis that the themes covered warranted the focus of the current paper.").

Our response

We appreciate the suggested guidance and have inserted the following text at the end of the section "Interview Schedule":

During the stepwise coding and analysis of the interviews, several themes concerning the challenges of treating this patient group emerged. This inductive process inspired an analysis where the psychiatrists' accounts were used and contextualised through a specific focus on their views of other HCPs' attitudes to DS and their ability and willingness to work with patients with DS.